# COCONut-PanCap: Joint Panoptic Segmentation and Grounded Captions for Fine-Grained Understanding and Generation

**Xueqing Deng**[1], **Linjie Yang**[1], **Qihang Yu**[1,2], **Ali Athar**[1,2], **Chenglin Yang**[1]
**Xiaojie Jin**[1,2], **Xiaohui Shen**[1], **Liang-Chieh Chen**[1,2]
[1]ByteDance Seed
[2]Work done while at ByteDance

## Abstract

This paper introduces the COCONut-PanCap dataset, created to enhance panoptic segmentation and grounded image captioning. Building upon the COCO dataset with advanced COCONut panoptic masks, this dataset aims to overcome limitations in existing image-text datasets that often lack detailed, scene-comprehensive descriptions. The COCONut-PanCap dataset incorporates fine-grained, region-level captions grounded in panoptic segmentation masks, ensuring consistency and improving the detail of generated captions. Through human-edited, densely annotated descriptions, COCONut-PanCap supports improved training of vision-language models (VLMs) for image understanding and generative models for text-to-image tasks. Experimental results demonstrate that COCONut-PanCap significantly boosts performance across understanding and generation tasks, offering complementary benefits to large-scale datasets. It establishes a new benchmark for evaluating models on joint panoptic segmentation and grounded captioning tasks, addressing the need for high-quality, detailed image-text annotations in multi-modal learning.

## 1 Introduction

Recent advancements in multi-modal foundation models have been largely driven by the availability of large-scale paired text-image datasets. These datasets, often collected via web crawling with basic filtering techniques [53, 54, 14], contain low-quality, web-sourced captions that lack depth and accuracy. In contrast, human-annotated caption datasets, such as COCO-caption [6], offer higher-quality descriptions but are limited in scale and tend to be concise, with an average caption length of 10 words. To overcome the limitations of short captions, the research community has leveraged vision-language models (VLMs) [39, 32, 33, 5, 61] to generate detailed synthetic captions. While these machine-generated captions improve visual understanding [33, 5] and generation tasks [32], they remain inferior to high-quality, human-verified annotations [45].

Addressing this challenge requires balancing scalability and annotation quality, as generating detailed and accurate image descriptions at scale remains labor-intensive [15, 45]. In this paper, we introduce an efficient annotation approach that combines dense mask annotations with commercial VLMs [5] to produce high-quality image captions. Our goal is to minimize human effort while generating rich, structured descriptions. To achieve this, we base our work on the COCO-caption dataset [6] due to its widespread use and diverse image content. We revisit the COCO-caption dataset to provide more detailed and comprehensive caption annotations. Our approach involves creating holistic captions synthesized from region-based dense captions that describe distinct areas within each image. Specifically, we build on recent COCONut panoptic segmentation annotations [9] to generate a new set of detailed captions by: (a) annotating each segmentation region with a VLM-generated

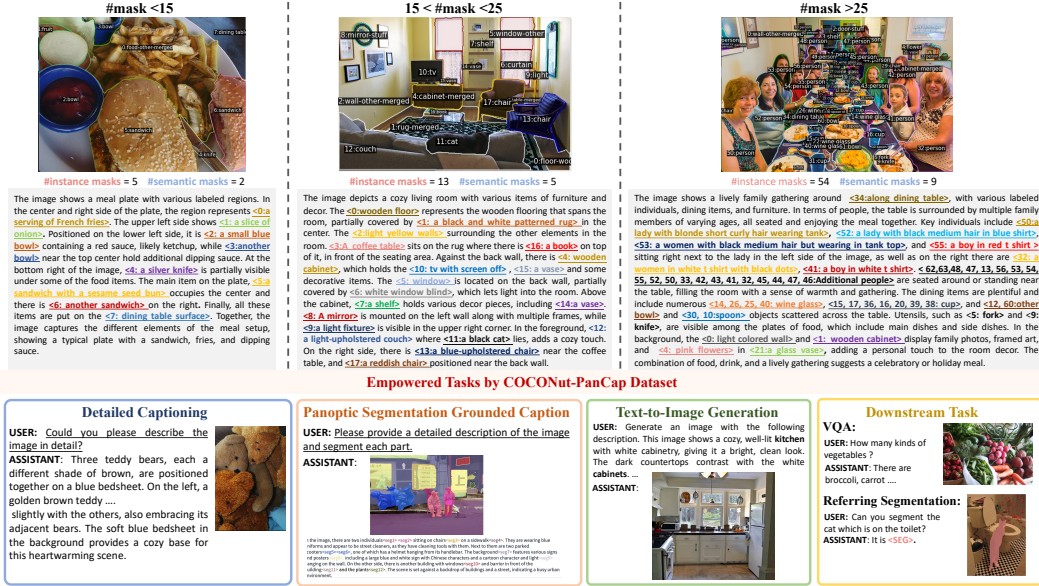

Figure 1: **COCONut-PanCap Dataset.** *Top:* The proposed COCONut-PanCap dataset features detailed captions grounded with dense panoptic segmentation masks. *Bottom:* COCONut-PanCap supports various fine-grained understanding and generation tasks, including detailed captioning, panoptic segmentation grounded caption, and text-to-image generation. The dataset also facilitates several downstream tasks, such as visual question-answering (VQA) and referring segmentation.

draft, carefully refined through human corrections, and (b) summarizing these region captions into a comprehensive image caption while preserving the grounding correspondence between image masks and object references. This enables a novel task that integrates panoptic segmentation with grounded captioning. Our structured annotation process ensures that the captions are both *complete*, covering the majority of objects in each image, and *grounded*, with precise segmentation masks.

The final dataset, named **COCONut-PanCap**, is designed for a wide range of vision-language applications, combining **Pan**optic segmentation and grounded **Cap**tioning. It comprises 118K image-text pairs for training, with an average caption length of 203 words, as well as an additional 25K image-text pairs, with an average caption length of 233 words for validation. We demonstrate that COCONut-PanCap significantly boosts the performance of both VLM and text-to-image generation models at the instruction tuning and fine-tuning stages, outperforming recent detailed caption datasets [45]. This highlights the potential of our grounding-based captions for both vision-language understanding and image generation tasks.

Our contributions are summarized as follows:

- We propose a caption annotation pipeline leveraging panoptic segmentation to create a high-quality, detailed caption dataset comprising **143K** (118K + 25K) annotated images. The resulting annotations are comprehensive, accurate, and include grounding masks, making this dataset substantially larger than recent detailed caption datasets.

- Our **COCONut-PanCap** dataset facilitates a new challenging task combining **P**anoptic segmentation and **G**rounded **C**aptioning (**PGC**). We establish evaluation metrics and settings for this PGC task and benchmark several recent methods to assess performance on this novel challenge.

- We validate the utility of our proposed dataset across various fine-grained Image-to-Text (I2T) and Text-to-Image (T2I) tasks, including detailed caption generation, PGC, text-conditioned image generation, visual question answering (VQA), and referring segmentation. Experimental results show that our dataset significantly enhances model performance across all these tasks.

Table 1: **Dataset (training set) Comparison.** Our proposed COCONut-PanCap dataset stands out for its **detailed** (2nd highest in Average Words), **high-quality** (human interactive annotation) *captions* and **high-density** *panoptic segmentation masks* (1st in Average Masks). ‡ denotes the mask number for referring segmentation which only counts the targets in QA format. Note that "Samples" means the number of collected annotations, where there may exist one image with multiple different annotations, *i.e.*, in region-level datasets like Osprey.

| Dataset Name | Image Source | Samples | Annotated by | Avg. Words | Masks |
|---|---|---|---|---|---|
| BLIP-LCS | LAION [54], CC [4], SBU [46] | 558K | BLIP [30] | 54 | ✗ |
| DenseFusion1M [33] | LAION [54] | 1,059K | Vision Specialist Models | 191 | ✗ |
| LLaVA-Recap118K [39] | COCO [36] | 118K | LLaVA-NEXT [39] | 186 | ✗ |
| LLaVA-Details-23K [38] | COCO [36] | 23K | GPT4 | 105 | ✗ |
| ShareGPT4V [5] | LAION [54], CC [4], SBU [46], COCO [36] *etc.* | 100K | GPT4-Vision | 162 | ✗ |
| ShareGPT4V-PT [5] | LAION [54], CC [4], SBU [46], COCO [36] *etc.* | 1,246K | Share-Captioner [5] | 144 | ✗ |
| PixelLM-MUSE [52] | LVIS [17] | 246K | GPT4-Vision | - | 3.7‡ |
| Osprey [71] | COCO [36] | 724K | GPT4-Vision | - | - |
| GLaMM-GCG [51] | RefCOCOg [41],PSG [68],Flick30K [48] | 214K | Vision Specialist Models | 128 | 3.6 |
| COCO-caption [6] | COCO [36] | 118K | **Human** | 11 | ✗ |
| DCI [62] | SA-1B [24] | 8K | **Human** | 144 | ✗ |
| DOCCI [45] | DOCCI [45] | 9.6K | **Human** | 136 | ✗ |
| IIW [15] | WebLI [15] | 8.5K | **Human** | 217 | ✗ |
| COCONut-PanCap (ours) | COCO [36] | **118K** | **Human** | 203 | **13.2** |

Table 2: **Dataset (evaluation set) Comparison.** Our COCONut-PanCap validation set provides detailed captions and supports multiple multi-modal tasks, including image captioning, text-to-image generation (T2I), and grounded segmentation (Grd. Seg.).

| Dataset Name | Samples | Avg. Words | Caption | T2I | Grd. Seg. |
|---|---|---|---|---|---|
| COCO-30K [6] | 30,000 | 11 | ✓ | ✓ | ✗ |
| DOCCI-test [45] | 5,000 | 136 | ✓ | ✓ | ✗ |
| IIW-test [15] | 445 | 217 | ✓ | ✓ | ✗ |
| GenEval [16] | 553 | 8 | ✗ | ✓ | ✗ |
| T2I-CompBench val [20] | 2400 | 9 | ✗ | ✓ | ✗ |
| GLaMM-GCG val-test [51] | 2,000 | 128 | ✓ | ✗ | ✓ |
| COCONut-PanCap val (ours) | 25,000 | 233 | ✓ | ✓ | ✓ |

## 2 Related Work

**Detailed Captions from VLMs.** Researchers are increasingly interested in creating large-scale datasets with detailed captions generated from advanced vision-language models. DenseFusion1M [33] utilizes a pretrained perceptual model to prompt VLMs, facilitating more detailed image descriptions. Recap-DataComp1B [32] first fine-tunes the Llama-3-8B powered LLaVA-1.5 model [37], then applies it to recaption approximately 1.3 billion images from the DataComp-1B dataset [14], generating a rich repository of detailed image descriptions. On a similar front, the PixelProse dataset [60] offers general-purpose image captions designed to serve various applications, from visual question answering (VQA) to pre-training tasks. Grasp Any Region [64] and DenseWorld1M [31] generate dense grounded captions by leveraging MLLMs. Unlike datasets targeting single applications, PixelProse captions are dense, versatile image descriptions that can be adapted to other formats, such as VQA and instructional data, with the help of large language models (LLMs). Although these detailed caption datasets are large-scale, they are directly generated by VLMs without human verification, falling behind human-annotated captions on quality. Our proposed COCONut-PanCap dataset leverages extensive human effort to ensure high-quality annotations.

**Human-annotated Detailed Captions.** Several efforts have been made toward this goal, utilizing fully human-annotated data or human-in-the-loop approaches. One example is DOCCI [45] which is a small, high-detailed image caption dataset that is entirely human-annotated, containing only 15K samples but providing diverse details, such as key objects, their attributes, spatial relationships, and text rendering. Two small-scale detailed caption datasets, ImageInWords [15] and DCI [62], use a combination of automatic annotation models with human involvement, both with fewer than 10K samples. Pixmo-Cap [8] introduces a large-scale dataset of detailed image captions from speech-based descriptions, offering richer visual annotations than text-based methods. Our proposed COCONut-PanCap dataset yields smaller scale compare to Pixmo-Cap but we have different focuses. Pixmo-Cap

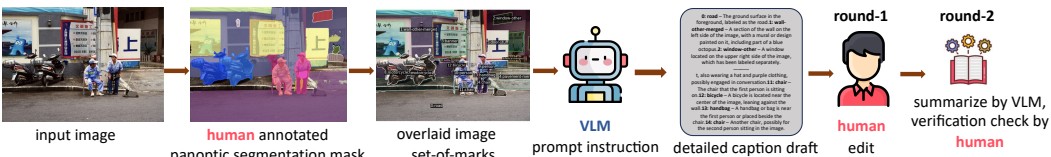

Figure 2: **Annotation Pipeline.** Given an input image, human-annotated panoptic segmentation masks are overlaid using set-of-marks [67] visualization techniques to prompt the vision-language model (VLM). After generating an initial draft, human effort is investigated for editing and verification. Finally, the annotated metadata will be formatted to construct the datasets for various tasks at instruction tuning or finetuning stage.

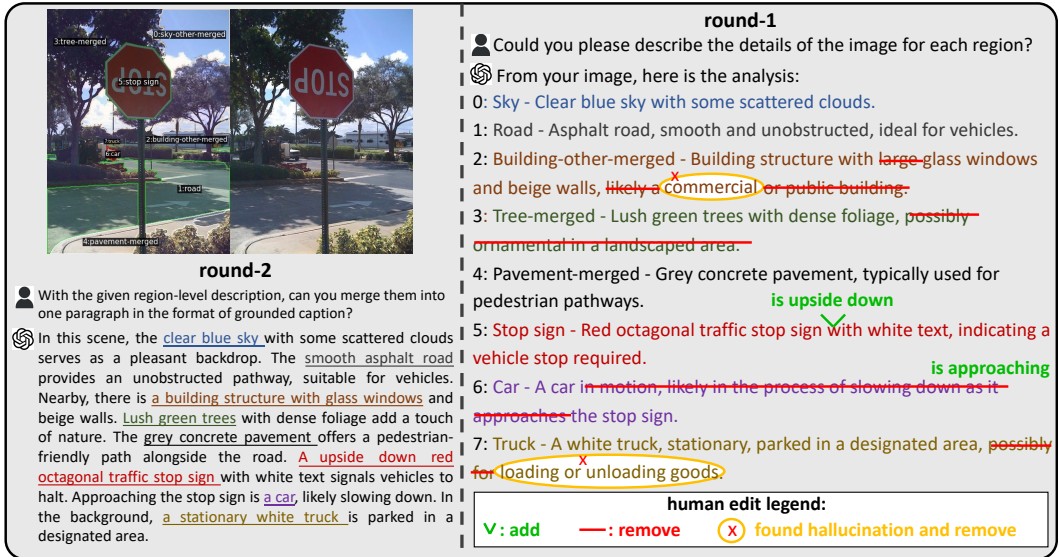

Figure 3: **Designed Prompt Template.** By giving the concatenated set-of-marks images, the right side (round-1) shows the initial response and the corresponding human edits. Once finalized by humans, these edits will be merged into a single detailed caption grounded with panoptic segmentation masks, as shown in the left side (round-2).

focuses on pretraining the VLMs, whereas we focus on the instruction tuning and finetuning stages of VLMs and image generation models. Our work also shares a similar annotation pipeline with a recent video captioning dataset Shot2Story [18], where both VLM draft and human corrections are used to create complete and accurate annotations.

**Grounded Captions with Segmentation Masks.** Existing work have made significant strides in creating datasets with region-level captions linked to entity segmentation masks [71] or bounding boxes [72]. However, few datasets associate grounded segmentation directly with captions. GLaMM [51] proposes a Grounding-anything Dataset (GranD) using an automated annotation pipeline that encompasses 7.5M unique concepts grounded in a total of 810M regions available with segmentation masks. Later, MGLMM [74] further explore the multi-granularity GLaMM model to generate a multi-granularity dataset. Our proposed COCONut-PanCap dataset follows a similar approach of grounding captions to dense masks but offers significantly denser masks per caption, as shown in Tab. 1, with an average of 13.2 masks per image compared to 3.6 in GLaMM. Note that we focus on grounded segmentation for detailed captions, rather than descriptions of all levels of segmentation masks (objects or parts) as provided in the GranD dataset [51], which is outside the scope of our study.

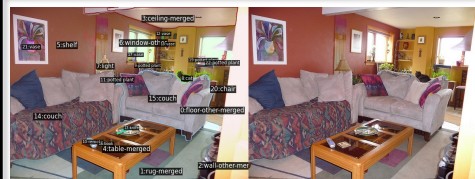 The image features a cozy and well-decorated living room. At the center of the room, <4:a wooden coffee table equipped with glasses> holds various items, including <10:a remote control>, <13: a knife> on the plates, and <16:a square small book>. On the left, The seating arrangement includes <14:a patterned couch with colorful cushions and blanket> and <15:another neutral-toned couch with vibrant throw pillows> , providing balance to the layout. The rug with colorful patters brings more warm atmosphere to the sitting area. Behind the couch, <20:A chair in the back> complements the seating options. Adding warmth to the room, <8:a black cat> rests comfortably on the couch. Behind the sitting area, there is <5:a 4-layer wall-mounted wooden shelf> with additional decorative items, including <11,12: vases> and other decorative items, enhancing the cozy and inviting atmosphere. Closed to the shelf, there are several <9,19,22:potted plants with green leaves > are placed throughout the room, adding a touch of greenery. <2:The wall painted in warm tones>, create a cozy atmosphere and are adorned with framed artwork and decorations. <0: The floor is neutral-toned>, supporting the entire setup. The <3:ceiling painted white>, contrasts subtly with the walls and reflects the natural light entering the room through <6:the large windows>.

Figure 4: **Visual Example of the Proposed COCONut-PanCap Dataset.**

## 3 *COCONut-PanCap* Dataset

We construct a novel dataset based on COCO images to provide detailed captions at both image and mask levels, using COCONut panoptic masks as a foundation for comprehensive region descriptions. Specifically, we leverage panoptic masks from COCONut-S [9] to annotate detailed region captions, incorporating both 'thing' and 'stuff' masks to cover a wide range of semantic regions.

### 3.1 Dataset Description

Comprehensively understanding diverse visual elements in complex scenes can benefit multiple tasks including perception, understanding, and generation. In this section, we describe the annotation pipeline for our dataset leveraging the human annotated panoptic masks. We first show the statistical analysis of our COCONut-PanCap in Tab. 1 (training set) and Tab. 2 (evaluation set). For training, our captions on average contain 203 words spanning 11 sentences along with 13.2 panoptic masks. We follow the same split setting in COCO2017 [36] dataset, which includes 118K training images. To provide a comprehensive evaluation set, we adopt the same 25K images from COCONut-val split [9], which contains COCO2017-val (5K images) and another 20K Objects365 [56] validation images.

### 3.2 Dataset Construction

We argue that high-quality descriptions should provide sufficient details of key objects and their attributes, as well as information about secondary objects and background elements. To achieve this, as shown in Fig. 2, we use human-annotated panoptic segmentation masks to decide the set of objects to reference in the caption. These masks include both 'thing' and 'stuff' classes, representing single objects and semantic regions, respectively. We adopt the panoptic segmentation masks from the COCONut-S [9] dataset. The masks are overlaid on the images, labeled with class names $c_1, c_2, \ldots, c_n \in C$, where $C$ is the set of COCO's 133 panoptic classes [36, 23]. We then construct a prompt with both the edited image and the original image, and a textual question for GPT-4V, as illustrated in Fig. 3. The resulting region captions from GPT-4V are reviewed and corrected by human raters for accuracy and consistency. The final annotations are illustrated in Fig. 1 and Fig. 4.

### 3.3 Dataset Analysis

**Concepts Beyond COCO's 133 Classes.** To clarify the goal of our annotation task, we focus on key visual features such as objects, attributes, spatial relationships, and counting. As shown in Fig. 5, we utilize the panoptic segmentation mask from COCONut-S, which includes 133 classes in the word vocabulary. Our proposed dataset, however, incorporates additional concepts beyond these 133 classes, such as 'vegetable' and 'parking'. This demonstrates that our human annotators delivers accurate and diverse descriptions when using the provided label names as a reference.

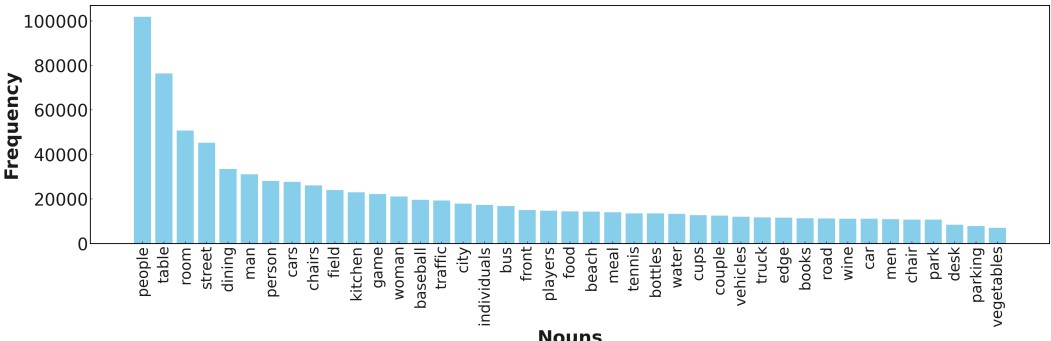

Figure 5: **Frequency of Extracted Nouns from the COCONut-PanCap Dataset**. The top 10 most frequent nouns are: people, table, room, street, dining, man, person, cars, chairs, and field.

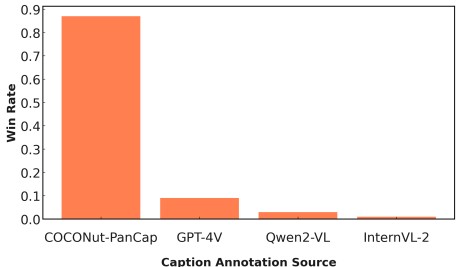

Figure 6: **Caption Quality via User Study.** The study involved human evaluators assessing a random sample of 1,000 captions, with a strong preference shown for captions from our dataset.

Figure 7: **Varying Synthetic and Human-Annotation Ratios.** CAPTURE is used to evaluate detailed captioning, while FID assesses the performance of image generation.

**User Study for Caption Quality.** We randomly sample 1,000 images from our COCONut-PanCap training set and asked a human evaluator to perform a single-choice selection task. The question is: *'Please select the best description for the image, considering the correctness of object names, attributes, counting, spatial relationships, and action.'* The compared captions are generated using GPT-4V [1], Qwen2-VL [66], and InternVL-2 [7], resulting in a single-choice four-option question. Fig. 6 illustrates the results, showing that our GPT-assisted human-annotated captions receives the highest ratings. More details can be found in the appendix.

## 4 Experimental Results

By utilizing COCONut-PanCap in the fine-tuning/instruction tuning stage, we assess its effectiveness by performing three primary tasks: detailed captioning, panoptic grounded captioning (PGC), and text-to-image generation. Additionally, we demonstrate the transferability of the knowledge learned from our dataset through two downstream tasks: VQA and referring segmentation.

**Primary Task 1: Detailed Captioning.** We conduct instruction tuning with LLaVA-NeXT framework [39] for this task. For a fair comparison, we replace the caption data (23k) from the original LLaVA instruction-tuning set with detailed captions from the same subset of our dataset, keeping the same amount of instruction data size. We follow the same training setup used for LLaVA-NeXT with Llama3-8B [11]. Treating it as a QA task, we use the prompt, *'Could you please describe the image in detail?'* and collect the corresponding response as the caption for the image. We evaluate caption quality using CIDEr [63], METEOR [2], BLEU@4 [47], ROUGE-L [35] and CAPTURE [10] metrics. For evaluating the captioning of dense objects, CAPTURE [10] provides accurate captioning metric. CAPTURE is a captioning evaluation metric designed to better align with human preferences by integrating referential similarity and factual consistency. It combines traditional n-gram overlap scores with image-text alignment signals derived from pretrained vision-language models, offering a more holistic assessment of caption quality.

Table 3: **Caption Benchmark Results Evaluated on Our COCONut-PanCap Val Set.** Note that the amount of data in the instruction dataset remains the same; only the sources of the detailed captions vary, with a total of 23K images that have detailed captions. * denotes reproduced results.

| Method | Pretrain Dataset | Instruction-tuning Dataset | Mask Pooled | CAPTURE | CIDEr | BLEU@4 | METEOR | ROUGE-L |
|---|---|---|---|---|---|---|---|---|
| LLaVA-NeXT* | LAION-CC-SBU | LLaVA 665K | ✗ | 55.4 | 10.8 | 4.2 | 13.2 | 23.1 |
| LLaVA-NeXT | LAION-CC-SBU | LLaVA 665K-ours | ✗ | 58.7 | 11.2 | 4.8 | 16.2 | 24.6 |
| LLaVA-NeXT-pool | LAION-CC-SBU | LLaVA 665K-ours | ✓ | 61.4 | 13.1 | 5.3 | 17.1 | 26.8 |
| LLaVA-NeXT-I | LAION-CC-SBU | LLaVA 665K-InternVL2-Cap | ✗ | 53.9 | 9.4 | 4.4 | 11.5 | 21.4 |
| LLaVA-NeXT-Q | LAION-CC-SBU | LLaVA 665K-Qwen2VL-Cap | ✗ | 55.4 | 8.9 | 4.6 | 12.9 | 22.5 |
| LLaVA-NeXT-G | LAION-CC-SBU | LLaVA 665K-GPT4V-Cap | ✗ | 56.2 | 9.6 | 4.7 | 13.3 | 22.8 |

Table 4: **Joint Panoptic Segmentation and Grounded Captioning (PGC) on COCONut-PanCap Val Set.** * denotes reproduced results.

| Method | Pretrain dataset | Instrt. dataset | Mask pooled | Caption | | | | Grounding segmentation | | |
|---|---|---|---|---|---|---|---|---|---|---|
| | | | | CAPTURE | CIDEr | BLEU@4 | METEOR | PQ | PQ$^{thing}$ | PQ$^{stuff}$ |
| LISA+ * | LAION-CC-SBU | GranDf | ✗ | 46.2 | 6.6 | 3.8 | 9.8 | 0.43 | 0.41 | 0.45 |
| LISA+ | LAION-CC-SBU | ours | ✗ | 57.9 | 8.1 | 4.9 | 13.8 | 0.50 | 0.49 | 0.44 |
| GLaMM GCG * | LAION-CC-SBU+GranD | GranDf | ✗ | 43.2 | 6.5 | 3.6 | 10.6 | 0.27 | 0.35 | 0.21 |
| GLaMM GCG | LAION-CC-SBU+GranD | ours | ✗ | 56.8 | 7.8 | 5.2 | 14.3 | 0.55 | 0.54 | 0.46 |
| PanCaper (ours) | LAION-CC-SBU | ours | ✗ | 62.6 | 12.0 | 5.8 | 15.4 | 0.56 | 0.55 | 0.66 |
| PanCaper-Pro (ours) | LAION-CC-SBU | ours | ✓ | 64.3 | 12.5 | 6.4 | 17.9 | 0.61 | 0.58 | 0.68 |

To enhance the dense object captioning ability, we also extend the model by adding the mask-pooled features from the panoptic segmentation masks as additional signals to the LLaVA model and name it LLaVA-NeXT-pool. During training, we use the ground truth mask to extract the features while during inference we use the mask proposals from the pretrained kMaX-DeepLab [69]. Besides, we also experiment with synthetic captions directly generated using InternVL-2 [7], Qwen2-VL [66] and GPT-4V [1]. We follow the same data preparation settings as our dataset to build these instruction datasets for these 23K images with different sources of synthetic detailed captions, namely LLaVA 665K-InternVL2-Cap , LLaVA 665K-Qwen2VL-Cap, and LLaVA 665K-GPT4V-Cap. These datasets are used to produce models LLaVA-NeXT-I, LLaVA-NeXT-Q, and LLaVA-NeXT-G, respectively.

The results are presented in Tab. 3. LLaVA-NeXT models show improved performance when fine-tuned on the custom instruction-tuning dataset (2nd row). Among these, LLaVA-NeXT-pool (3rd row) achieves the highest scores in all metrics, with CAPTURE of 61.4, CIDEr of 13.1, BLEU@4 of 5.3, and METEOR of 17.1, significantly higher than the original model variant LLaVA-NeXT (1st row), indicating the benefit of added region features for additional visual cues. Models trained on synthetic captions (LLaVA-NeXT-I, LLaVA-NeXT-Q, and LLaVA-NeXT-G) generally show lower scores, showing advantage of our human-annotated caption.

**Primary Task 2: A New Pixel-Grounding Task (PGC) with Stronger Detail Reasoning Performance.** The proposed COCONut-PanCap dataset enables a new pixel-grounding task: Joint Panoptic Segmentation and Grounded Captioning (PGC). To build a strong baseline on this challenging task, we develop PanCaper based on LISA [28] which uses pre-trained LLaVA-NeXT with Llama3-8B (and fine-tuned with LoRA [19]). The vision encoder uses a fixed CLIP-ViT-L/14-336 model, modified with linearly interpolated position embeddings to process 448 resolution images. The trainable components of our model include the mask decoder of kMaX-DeepLab, and the tunable parts in LLaVA are the same as in LISA. To enhance model performance in visual understanding, we initialize our PanCaper using pretrained LLaVA-NeXT models from the detailed captioning task. We also experiment with a model variant that uses mask pooled features similar to LLaVA-NeXT-pool, and name it PanCaper-Pro. We provide more details of PanCaper in the appendix.

For comparison, we select 3 related methods LISA, PixelLM [52] and GLaMM [51] for evaluation. It is noteworthy that LISA is not able to perform multi-mask prediction. We therefore adapt LISA [28] for the multi-mask generation with grounded segmentation, namely LISA+. We introduce a benchmarking suite for the PGC task, with a validation set of 25K images. For the caption quality, we report the caption metrics including CIDEr [63], METEOR [2], ROUGE-L [35], BLEU@4 [47] and CAPTURE [10]. For grounded panoptic segmentation, we report PQ scores [23]. Tab. 4 shows the quantitative results. Our proposed PanCaper-Pro achieves the highest scores across all captioning metrics (CIDEr: 12.5, CAPTURE: 64.3, BLEU@4: 6.4, METEOR: 17.9), outperforming all other models. Both PanCaper models show significant improvements over other models in all captioning metrics, highlighting the effectiveness of the COCONut-PanCap dataset for detailed

Table 5: **Benchmark Results on Text Conditioned Image Generation.** Stable-Diffusion-3 (SD3) medium is finetuned with COCO-Caption (short), DOCCI and our COCONut-Panoptic and evaluated on DOCCI test set [45] and our COCONut-PanCap val set. 'SD3 PT dataset' denotes the pretraining dataset of SD3, and thus the rows correspond to zero-shot evaluation of SD3.

| Training dataset | Evaluation dataset | FID↓ | FD$_{dinov2}$ ↓ | CLIPScore↑ |
|---|---|---|---|---|
| SD3 PT dataset [12] | | 30.2 | 345 | 74.9 |
| COCO-caption [6] | DOCCI test set [45] | 27.6 | 321 | 76.8 |
| DOCCI [45] | | 22.1 | 300 | 77.8 |
| COCONut-PanCap (ours) | | 21.4 | 290 | 77.9 |
| SD3 PT dataset [12] | | 31.8 | 300 | 73.8 |
| COCO-caption [6] | COCONut-PanCap | 28.0 | 294 | 74.0 |
| DOCCI [45] | val set (ours) | 24.3 | 267 | 75.1 |
| COCONut-PanCap (ours) | | 23.1 | 260 | 77.3 |

Table 6: **Effects of Fine-tuning the SD3-medium (T2I model) with Different Datasets on GenEval [16].** 'w/o FT' denotes the model is not finetuned (*i.e.*, zero-shot testing).

| | w/o FT | COCO-caption [6] | DOCCI [45] | COCONut-PanCap |
|---|---|---|---|---|
| color attribution | 0.37 | 0.34 | 0.38 | 0.40 |
| colors | 0.73 | 0.70 | 0.74 | 0.75 |
| position | 0.33 | 0.30 | 0.36 | 0.36 |
| counting | 0.65 | 0.64 | 0.65 | 0.70 |
| single object | 0.96 | 0.94 | 0.95 | 0.96 |
| two objects | 0.80 | 0.78 | 0.81 | 0.89 |
| overall score | 0.64 | 0.62 | 0.65 | 0.68 |

caption generation. On grounding segmentation, PanCaper-Pro again leads, with a PQ score of 0.61, PQ$^{thing}$ of 0.58, and PQ$^{stuff}$ of 0.68, reflecting its robustness on both 'thing' and 'stuff' classes. Notably, enabling mask pooling in our proposed PanCaper-Pro further enhances segmentation metrics. The baseline models (LISA+ and GLaMM with GranD) achieve much lower PQ scores, due to incomplete segmentation annotations in the GranD dataset.

**Primary Task 3: Text-to-Image Generation.** We adopt the Stable Diffusion 3 (SD3) medium model[1] for text to image generation with LoRA finetuning. We adopt the default training settings but only with different text-image datasets for training. We evaluate with two types of training images from COCO [36] and DOCCI [45] datasets. In details, for the COCO images, we explore the short COCO-caption and detailed captions from our dataset. For DOCCI images, we directly use the captions from their dataset. Tab. 5 shows the quantitative results. Traning on COCONut-PanCap achieves the best performance across all metrics when evaluated on DOCCI-test, with the lowest FID (21.4), lowest FD$_{dinov2}$ (290), and the highest CLIPScore (77.9), indicating superior generation quality and high image-text relevance. When evaluated on COCONut-PanCap-val set, training on COCONut-PanCap again shows the best results with the lowest FID (23.1), FD$_{dinov2}$ (267), and a high CLIPScore of 77.3.

Tab. 6 shows the results on GenEval benchmark [16]. Finetuning SD3-medium with COCONut-PanCap consistently scores the highest in most categories, particularly those requiring image details like color attribution, object positioning, and handling multiple objects. Our proposed dataset enables more accurate image generation that requires understanding of relationships, multiple objects and counting, tasks that other datasets struggle with.

**Downstream Task 1: VQA.** To evaluate the effectiveness of COCONut-PanCap dataset, we utilize the captions during the instruction-tuning stage and follow the setup of LLaVA-NeXT [39] across various visual question answering (VQA) and multi-modality understanding benchmarks. We evaluate on MM-Vet [70], SEED-IMG [29], MMBench-en [40], MME [13], POPE [34], and TextVQA [59], covering a broad range of evaluation dimensions. We experiment with different amount of our COCONut-PanCap caption data injected into the instruction tuning stage by replacing the original COCO captioning data with our dataset. As shown in Tab. 7, the baseline model LLaVA-NeXT (using its original recaptioned COCO) achieves relatively lower performance across all metrics, with scores

---

[1]https://huggingface.co/docs/diffusers/stable_diffusion/stable_diffusion_3

Table 7: **Benchmark Results and Ablation Study on VQA.** By adding extra detailed caption data for instruction tuning, the models show increased improvement. * denotes reproduced results. Using only **20K human labeled data** can still achieve **comparable performance** to 100K synthetic data.

| Method | LLM | Instruction-tuning Dataset | MM-Vet | Seed-IMG | MMBench-en | TextVQA | POPE | MME |
|---|---|---|---|---|---|---|---|---|
| LLaVA-NeXT * | Llama3-8B | orginal LLaVA 665K [39] | 43.5 | 70.1 | 71.4 | 68.9 | 85.4 | 1523 |
| LLaVA-NeXT-20K | Llama3-8B | LLaVA 665K-COCONut-PanCap-20K | 44.1 | 72.5 | 73.6 | 69.8 | 86.1 | 1552 |
| LLaVA-NeXT-50K | Llama3-8B | LLaVA 665K-COCONut-PanCap-50K | 44.6 | 73.1 | 74.2 | 70.0 | 87.1 | 1600 |
| LLaVA-NeXT-Full | Llama3-8B | LLaVA 665K-COCONut-PanCap-118K | 45.5 | 74.3 | 75.1 | 70.7 | 87.9 | 1612 |
| LLaVA-1.5 | Vicuna-7B | LLaVA 665K-ShareGPT4V-100K | 37.8 | 67.4 | 70.5 | 64.6 | 84.7 | 1519 |
| LLaVA-1.5 | Vicuna-7B | LLaVA 665K-COCONut-PanCap-20K | 38.5 | 67.7 | 70.9 | 64.5 | 84.9 | 1521 |

Table 8: **Benchmark Results on Referring Segmentation.** * denotes reproduced results. It is noted that GLaMM uses extra data from the GranD dataset for pretraining. $^+$ denotes our PanCaper model is adapted for referring segmentation task.

| Method | refCOCO | | | refCOCO+ | | | refCOCOg | |
|---|---|---|---|---|---|---|---|---|
| | val | testA | testB | val | testA | testB | val | test |
| GLaMM* [51] | 77.5 | 79.2 | 74.9 | 71.3 | 74.7 | 61.5 | 71.3 | 71.9 |
| PixelLM [52] | 73.0 | 76.5 | 68.2 | 66.3 | 71.7 | 58.3 | 69.3 | 70.5 |
| LISA-7B [28] | 74.1 | 76.5 | 71.1 | 62.4 | 67.4 | 56.5 | 66.4 | 68.5 |
| PanCaper$^+$ | 74.5 | 76.7 | 69.9 | 69.9 | 73.4 | 59.5 | 69.8 | 70.6 |
| PanCaper$^+$ + COCONut-PanCap | 76.2 | 77.1 | 72.3 | 70.5 | 73.9 | 60.1 | 72.1 | 71.6 |

such as 43.5 on MM-Vet, 70.1 on Seed-IMG, and 68.9 on TextVQA. Building on LLaVA-NeXT baseline, we progressively incorporated varying amounts of COCONut-PanCap data (20K, 50K, and 118K (full), as indicated by postfixes in the baseline names) during instruction-tuning. Consistent improvements are observed across all evaluated benchmarks as more of our data is integrated.

**Downstream Task 2: Referring Segmentation.** In this task, the model processes an image and a textual referring expression to output a segmentation mask corresponding to the expression. The prompt used is, *'Please segment the* ⟨referring_text⟩ *in the image.'* The target model response is *'Sure, it is* ⟨SEG⟩*.'*, where the ⟨SEG⟩ token is decoded to obtain the mask. We follow the setup in LISA [28], using multiple segmentation datasets to jointly train the models. Tab. 8 shows the quantitative results. Our model achieves superior performance, particularly when additionally trained with the COCONut-PanCap dataset (last row), outperforming all models except GLaMM [51]. This improvement underscores our model's efficacy in handling complex referring expressions, likely due to the additional data that enhances model generalization and accuracy. It is worth noting that GLaMM performs competitively with our method, though the comparison is uneven given their additional use of the SA-1B dataset [25].

**Discussion: Synthetic *vs*. Human Annotated Data.** Generating synthetic data for captioning has been popular for recent tasks in either training vision encoders [49] or text-to-image generation [32]. Therefore, we investigate the effect of varying the mix ratio of synthetic captions generated by GPT-4V and our human-annotated data for fine-tuning in Fig. 7 (where x-value 0.0 indicates fully synthetic data), using the COCONut-PanCap dataset for training and the COCONut-PanCap-val set for evaluation. We adopt LLaVA-NeXT for the captioning task and SD3-medium for the image generation task. As shown in the figure, adding 25% human-annotated data yields significant performance improvements in both captioning and generation, with a reduced FID of 26 from 31 (lower is better) and an increased CAPTURE score of 53.6 from 47.5 (higher is better). Consistent improvements are observed as more human-annotated data is incorporated.

# 5 Conclusion

In this work, we proposed a novel dataset designed to support detailed captioning and grounded segmentation tasks built on COCO images. We demonstrated that our dataset can enhance model performance during instruction tuning and fine-tuning stages across various multi-modal understanding and generation tasks, such as captioning, grounded segmentation, and text-to-image generation. We hope that COCONut-PanCap, with its detailed captions grounded with dense panoptic masks, will foster future advancements in multi-modal learning research.

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

## A   Experimental Details

In this section, we provide more implementation details for detailed captioning (primary task 1) in Sec. A.1, PGC (primary task 2) in Sec. A.2, and VQA (downstream task 1) in Sec. A.3.

### A.1   Detailed Captioning

**Detailed Captioning Instruction Dataset Construction.** The key step in conducting the experiment is constructing the dataset. The original LLaVA-665K dataset consists of LLaVA-158K combined with other VQA datasets. Within LLaVA-158K, a subset of detailed captions corresponds to 23K COCO images. To create our-LLaVA-665K (referred to as LLaVA 665K-COCONut-PanCap in the table), we replace the detailed caption annotations for these 23K COCO images with our annotations. Importantly, the total amount of training data remains unchanged (only the captions for these 23K images are updated), ensuring a fair comparison of the impact of data quality on model performance. To train LLaVA-NeXT on detailed captioning, we use 16×A100-40G to conduct the experiment and the training time is around 8 hours.

**Synthetic Annotation for Detailed Caption.** To build the synthetic dataset with state-of-the-art VLM, we use three models, including open-sourced InterVL-2, Qwen2-VL and close-sourced GPT-4V to generate the detailed captions for COCO 118K train set images. We use the same text prompts that is used in LLaVA [38] for prompting the model to create the detailed captions.

**LLaVA-NeXT-pool implementation details.** Fig. 8 shows the comparison of the original LLaVA-NeXT and our proposed LLaVA-NeXT-pool. As shown in Fig. 8a, in order to preserve the details for the high-resolution images and representations, the original design employs a grid configuration which can also balance the performance efficiency with operational costs. Then both the patch-level and image-level features are later concatenated and sent to the LLM. Directly splitting the image into patches could cause prolems, for example, in the figure, the upper part of the dog's head is partitioned into different patches which may result in incomplete feature extraction for single object. To overcome this drawback, we propose LLaVA-NeXT-pool to extract the dense feature and preserve the object details by utilizing the panoptic segmentation masks in our COCONut-PanCap dataset. Fig. 8b shows the details. Compared to the original design, LLaVA-NeXT-pool could effectively extract the features for the dog in our example. Our design enables more complete region-level feature extraction and is potential in understanding the details better.

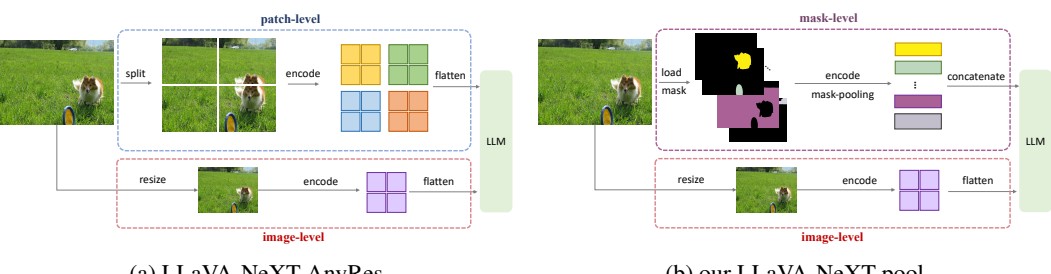

(a) LLaVA-NeXT-AnyRes                    (b) our LLaVA-NeXT-pool

Figure 8: **Comparison of LLaVA-NeXT and our proposed LLaVA-NeXT-pool.**

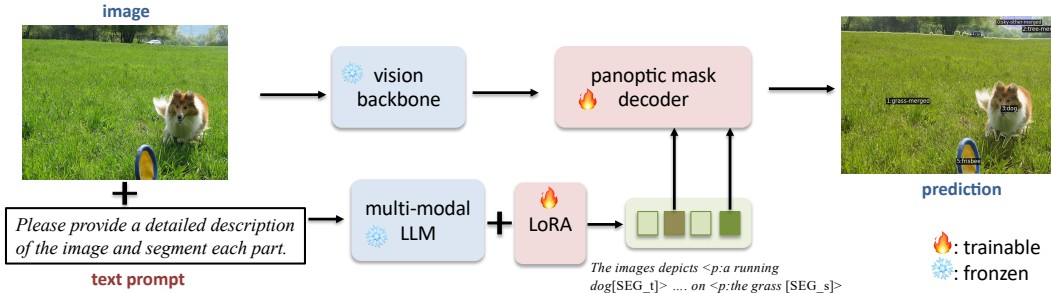

Figure 9: **Architecture of PanCaper.** We utilize a pretrained vision encoder from kMaX-DeepLab [69] as our vision backbone, which effectively extracts dense features essential for panoptic segmentation.

## A.2 A New Task: Joint Panoptic Segmentation and Grounded Captioning (PGC)

In this section, we introduce our baseline method for joint panoptic segmentation and grounded captioning (PGC), namely PanCaper. We start with an overview of the pixel grounding task and then present our proposed approach, which incorporates a panoptic segmentation module specifically designed for grounding objects in captions.

**Revisiting the Pixel Grounding Task.** Our baseline model builds upon LISA [28], a model that combines the language generation capabilities of VLMs with the ability to produce segmentation mask. LISA consists of three main components: a VLM, a vision backbone $V$, and a mask decoder $D$. With a given text prompt, the VLM (typically LLaVA [38, 37]) generates an output containing a $\langle \mathrm{SEG} \rangle$ token. For instance, with the input prompt, *'Could you segment the food with high Vitamin C?'* LISA generates the response *'It is* $\langle \mathrm{SEG} \rangle$.' This process extracts the last-layer embedding of the LLM from LLaVA. Then a language-to-prompt (L-P) projection layer ($g$) transforms the last-layer embeddings corresponding to $\langle \mathrm{SEG} \rangle$ tokens ($l_{\mathrm{seg}}$) into the decoder's feature space. Meanwhile, the vision backbone extracts dense visual features from the input image. Finally, both the dense features and the CLIP image embedding from LLaVA are fed into the mask decoder to produce the final segmentation mask.

**Prompt Instruction for Grounded Captioning.** We propose a baseline method for the PGC task by modifying LISA to enable grounded captioning with segmentation masks. Since LISA was originally designed for generating segmentation with a single output mask, two main adjustments are necessary: (1) the use of multiple $\langle \mathrm{SEG} \rangle$ tokens, and (2) extracting noun phrases from the caption for grounding. To facilitate grounded segmentation, we modify the prompt to the VLM as *'Please provide a detailed description of the image and segment each part.'* This prompt triggers the model to generate caption responses with corresponding $\langle \mathrm{SEG_i} \rangle$ tokens, where $i \in [1, N]$ and $N$ is the total number of predicted segmentations. Given a predicted caption for the image, aligning each $\langle \mathrm{SEG_i} \rangle$ token requires pairing it with a noun phrase, '$\langle \mathrm{p} \rangle \mathrm{phrase_i} \langle /\mathrm{p} \rangle$,' where $\mathrm{phrase_i}$ is the relevant part in the caption to be grounded. With these prompt tokens defined, the model uses the vision backbone $V$ and mask decoder $D$ to facilitate fine-grained, pixel-level grounding, with $D$ producing segmentation masks $M$.

**Adapting Baseline Methods for PGC Task.** We adopt the same text prompt template to enable the model to perform PGC tasks. For LISA+, we follow the same design in GLaMM [51] to design the multi entity mask output by utilizing the the GranDf dataset. As the intruction dataset of GranDf is constructed similarly grounding the phrase in the image-level caption, it will output multiple $\langle \mathrm{SEG} \rangle$ tokens. The reasoning results of the number of $\langle \mathrm{SEG} \rangle$ tokens decide the number of output entity mask which are often binary masks. As a result, the model can generate a detailed caption along with interleaved segmentation masks, employing the format "$\langle \mathrm{p} \rangle$A man$\langle /\mathrm{p} \rangle \langle \mathrm{SEG} \rangle$ ... next to $\langle \mathrm{p} \rangle$a tree$\langle /\mathrm{p} \rangle \langle \mathrm{SEG} \rangle$". And thus the format of instruction dataset is significat in task design. Therefore, we formulate our dataset as "$\langle \mathrm{p} \rangle$A man$\langle /\mathrm{p} \rangle \langle \mathrm{SEG_t} \rangle$ ... next to $\langle \mathrm{p} \rangle$a tree$\langle /\mathrm{p} \rangle \langle \mathrm{SEG_s} \rangle$", where $\langle \mathrm{SEG_t} \rangle$ represents the seg token for instance masks of thing and $\langle \mathrm{SEG_s} \rangle$ represents for semantic masks of stuff respectively in panoptic setting. Similarly, utilizing the PanCap dataset and special token design, GLaMM [51] is able to generate the entity masks with the tag of 'thing' and 'stuff'.

**Enable Panoptic Grounding**. To achieve panoptic segmentation from captions, we first classify $\langle \text{SEG} \rangle$ tokens into two types: $\langle \text{SEG}_t \rangle$ for 'thing' classes and $\langle \text{SEG}_s \rangle$ for 'stuff' classes. These tokens are then processed by our segmentation modules to produce panoptic segmentation masks. We initialize the vision backbone $V$ with a pretrained kMaX-DeepLab encoder [69] and fine-tune the decoder $D$ using our COCONut-PanCap dataset. Since kMaX-DeepLab operates as a closed-set segmenter, we align text embeddings of the associated noun phrases with COCO's 133 panoptic classes. To accomplish this alignment, we use BERT [26] to generate the text embeddings and to calculate cosine similarity, selecting the best-matching category. Panoptic grounding provides mapping between detailed captions and image regions, which improves interpretability of VLM predictions.

**Training Objectives.** Our training objective aims to minimize the following losses:

$$\mathcal{L} = \lambda_{\text{text}}\mathcal{L}_{\text{text}} + \lambda_{\text{mask}}\mathcal{L}_{\text{mask}}, \tag{1}$$

where $L_{\text{text}}$ is the auto-regressive cross-entropy loss for text generation, and $L_{\text{mask}}$ is the mask loss [65], encouraging the model to produce high-quality segmentation results. $\lambda_{\text{text}}$ and $\lambda_{\text{mask}}$ are the respective loss weights. We use the same loss weights as LISA [28].

**Training Data Formulation.** We adopt the same training data from LISA [28] which comprises mainly three parts, all of which are derived from widely-used public datasets. These include 1) Semantic Segmentation datasets including ADE20K [73], COCO-Stuff [3], and LVIS-PACO [50] part datasets with the generated QA data, 2) Vanilla Referring Segmentation Datasets: refCOCO, refCOCO+, refCLEF [22] and refCOCOg [41] datasets, 3) ReasonSeg dataset [28], and 4) Visual Question Answering Dataset: LLaVA-v1.5-mix665k [37]. To enable the multi-mask generation for grounded caption, there are two options for instruction datasets, GranDf and our COCONut-PanCap where GranDf consists of entity masks while COCONut-PanCap consists of panoptic masks.

**Evaluation Metrics for Caption Quality.** We conduct the analysis with multiple metrics to evaluate the quality and completeness of the generated captions. We introduce a benchmarking suite for the PGC task, with a validation set of 25K images. For the caption quality, we report the caption metrics including CIDEr [63], METEOR [2], ROUGE-L [35], BLEU@4 [47] and CAPTURE [10]. For grounded panoptic segmentation, we report PQ scores [23].

**PanCaper Implementation Details.** Following the architecture in LISA [28], there are three components including the vision backbone, mask decoder and multi-modal LLM. Fig. 9 shows the architecture details for PanCaper. We made modification on the vision backbone, and mask decoder part in terms of model architecture. To preserve the learned knowledge of the pre-trained multimodal LLM (*i.e.*, LLaVA-NeXT in our experiments), we leverage LoRA [19] to perform efficient fine-tuning, and completely freeze the vision backbone. The mask decoder is fully fine-tuned. Additionally, the LLM token embeddings (embed tokens), the LLM head (lm head), and the projection layer are also trainable. The weights of the text generation loss $\lambda_{\text{text}}$ and the mask loss $\lambda_{\text{mask}}$ are set to 1.0 and 1.0, respectively. For the PQ-style mask loss, we follow the same settings in kMaX-DeepLab [69], where it consists of mask-level cross entropy loss, dice loss and pixel loss. The training takes 10 hours with 8 A100-40G.

## A.3 VQA

We provide more implementation details for the VQA experiments. We follow the same setting in LLaVA-NeXT to create the experimental results for VQA tasks. We focus on the instruction tuning stage by adopting the pretrained weights from the stage-1 across the trainings for all the model variants mentioned in Tab. 7 in the paper. The dataset we used is exactly the same as in LLaVA 665K [37] which includes the earlier version of instruction data proposed in LLaVA 158K [38], ShareGPT [57], VQAv2 [42], GQA [21], openknowledge VQA (OKVQA [43], A-OKVQA [55]), OCR (OCRVQA [44], TextCaps [58]), region-level VQA datasets (Visual Genome [27], RefCOCO [22]). Among these data, LLaVA 158K comprises 77K complex reasoning, 58K conversation and 23K detailed captions. To build the dataset variants shown in Tab. 7, we simply remove the subset of detailed_caption_23k, and subsequently add 20K, 50K and 118K COCONut-PanCap dataset to build LLaVA 665K-COCONut-PanCap-20K, LLaVA 665K-COCONut-PanCap-50K and LLaVA 665K-COCONut-PanCap-118K. By these steps, we add more detailed caption data to construct the instruction tuning dataset. This results in the total amount of training data of 662K for LLaVA 665K-COCONut-PanCap-20K, 692K for LLaVA 665K-COCONut-PanCap-50K

and 760K for LLaVA 665K-COCONut-PanCap-118K. And thus the size of LLaVA 665K-COCONut-PanCap-20K is slightly smaller than the original LLaVA 665K dataset, but the model trained on it yields better performance. For the evaluation settings, we follow the exact settings in LLaVA-NeXT [39] using lmms_eval[2]. For the stage-1 training, it takes 8 A100-40G to train for around 12 hours.

## B  More Qualitative Results

In this section, we present additional qualitative results of COCONut-PanCap annotations (Sec. B.1) and a detailed analysis of tier cases from the user study (Sec. B.2).

### B.1  Data Examples

We show more visualization of our proposed COCONut-PanCap dataset in Fig. 10 and Fig. 11.

### B.2  PanCaper and GPT-4V Tier Showcases

In the user study involving 1,000 samples, captions generated by GPT-4V were preferred in 87 cases. Among these, actually, 46 were tier cases where human raters considered both GPT-4V and COCONut-PanCap captions equally good. Fig. 12, Fig. 13 and Fig. 14 illustrate qualitative examples, highlighting the reasons for the tier classification and instances where GPT-4V was chosen.

## C  Limitations

High-quality human-labeled data offers significant benefits for instruction tuning in multi-modal tasks, but scaling such datasets is challenging. To address this, we introduce COCONut-PanCap as a starting point for large-scale human-annotated data exploration. Recognizing the relatively smaller dataset size compared to other large dataset, future work may involve using this dataset to train seed models to generate more high-quality synthetic data.

## D  Broader Impact

The COCONut-PanCap dataset is designed to advance research in vision-language understanding and generation by providing fine-grained, panoptic-grounded captions. The dataset is intended to support the development of more precise, coherent, and semantically grounded vision-language models, which can have a wide range of beneficial societal applications.

**Positive Societal Impacts.** COCONut-PanCap has the potential to significantly improve downstream applications in accessibility (e.g., more descriptive image captions for visually impaired users), robotics (e.g., fine-grained scene understanding for autonomous agents), education (e.g., AI-assisted visual learning tools), and scientific analysis (e.g., detailed visual documentation). By grounding captions in panoptic segmentation, the dataset encourages models to produce more accurate and interpretable outputs, which can enhance trustworthiness and explainability in AI systems.

**Negative Societal Impacts.** As with other datasets that improve image-text alignment and generation, there are potential risks associated with misuse. Enhanced image captioning and generation capabilities may contribute to the creation of misleading or deceptive content, such as realistic fake captions or synthetic images used in misinformation or manipulation. We commit to continuously monitoring the usage of our released models and codes and will take action to restrict access or provide additional guidance if we identify concerning patterns of misuse.

In summary, while COCONut-PanCap aims to push forward the state of multimodal understanding and generation, we acknowledge the dual-use nature of such technologies and advocate for responsible development and deployment practices.

---

[2]https://github.com/EvolvingLMMs-Lab/lmms-eval

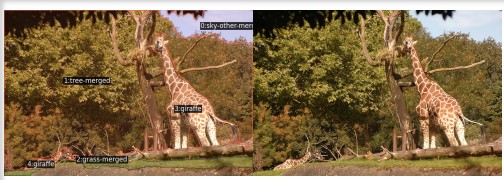

The image depicts a natural outdoor with trees and giraffes. <0:The sky is blue>, forming the backdrop of the scene. Below it, **there are** <1:dense trees, filled with branches and lush green leaves>. Within this environment, two giraffes are prominently featured. The image mainly focuses on <3:a standing giraffe with a long neck and unique patterns> , actively eating leaves from the tree. In contrast, there is <4: a second giraffe with similar distinctive patterns>, which is far away from the previous giraffe is resting comfortably on the grass. Both giraffes are surrounded by green trees and <2:a grassy area, predominantly covered with green grass interspersed with patches of exposed brown soil>.

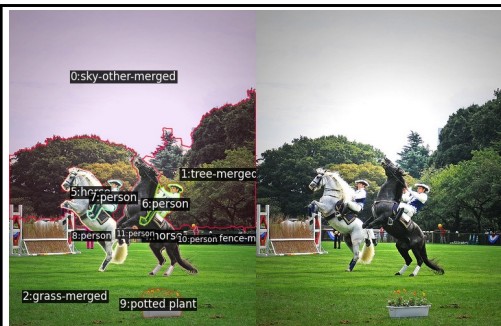

The image depicts a dynamic outdoor scene where people are riding horses. In the foreground, two horses take center stage. <9:A black horse with a white mane and tail behind its neck, adorned with a brown bridle, a predominantly dark blue saddle with yellow patterns, and blue leg wraps>, is raising its front hoof. Beside it, <5:a white horse with a black mane and tail, wearing a black bridle, a similar dark blue saddle with yellow patterns, and white leg wraps>, is also raising its front hoof. Both horses are being controlled by <6,7: two man who are dressed in blue and white tops, white pants, and black boots>, actively taming the horses. They are riding horses on <2:a vivid green grassland> that provides the base for the action. Adding structure, there is <3: a fence made of wooden posts and railings> in the background. There are some people in the background that are obscured by the horses. For example, there is <8:a person wearing black pants> partially obscured by the white horse and <10:another person in a red top and white pants> who is watching the activity; and <11:a person in a red top and black pants>, partially hidden by the black horse. Together, the elements create a cohesive portrayal of a lively horse-taming event set against a serene natural background. The weather is nice, as <0:the sky is white and cloudless>, forming the backdrop. Below it, there features <1:a dense cluster of trees with brown trunks and green leaves>, framing the scene.

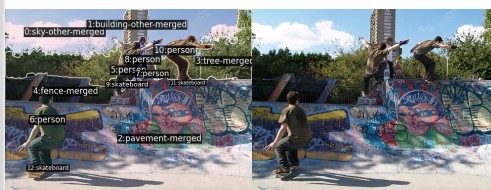

The scene includes several individuals actively engaging in skateboarding. There are <6,10: two boys> wearing in green top and black pants>, actively playing <9:11skateboard> in the air. <5: Another guy who is also dressing in green top and black pants> is playing but on the ground. Next to them, there are <7: a half-naked man> observing the skateboarding performance, while <8: another guy in a white top and black hat>, also watching the activities. Skateboards are prominently featured in the center area, which includes a black skateboard deck used for tricks. Lastly, the background shows the <0:sky, predominantly blue with scattered clouds>. <1:A light brown building> is obviously seen in the background. The skaters are using the <2:sidewalk, notable for its graffiti and colorful markings>. Around the scene, there is <3:lush green foliage>, adding natural scenery to the skate park.

Figure 10: **Visualization of the Panoptic Grounded Caption.** Our annotated captions ground the panoptic segmentation masks.

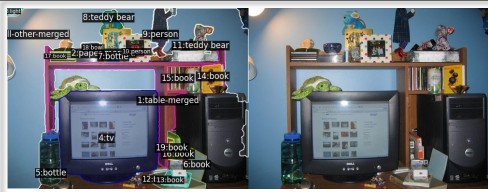

This image showcases a well-organized desk setup. On <1:the wooden desk with a shelf>, there is <4:a DELL computer> occupies the central space, displaying content on its screen. Besides, there is a turtle toy on top of it. Surrounding the computer, multiple items are neatly arranged. To its left, <5:a blue water bottle> stands prominently, next to <6:a book> lying on the desk. Below the computer, <12,13,16,19:additional books> are placed. On the upper shelves, various objects add character to the space like <14,15: books> and a drink can. At the top of the shelves, <8:a fluffy blue teddy bear > is positioned on the left, and <11:another teddy bear > is positioned on the right, adding a playful touch. There are various small items as well, like <7:a glass bottle>, <17:books> and a photo frame. In the background, <0:the wall is painted blue>, serving as the backdrop for the scene.

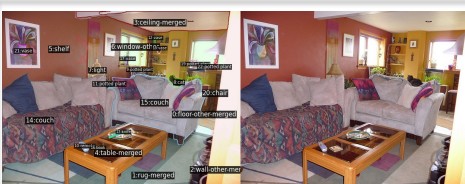

The image features a cozy and well-decorated living room. At the center of the room, <4:a wooden coffee table equipped with glasses> holds various items, including <10:a remote control>, <13: a knife> on the plates, and <16:a square small book>. On the left, The seating arrangement includes <14:a patterned couch with colorful cushions and blanket> and <15:another neutral-toned couch with vibrant throw pillows> , providing balance to the layout. The rug with colorful patters brings more warm atmosphere to the sitting area. Behind the couch, <20:A chair in the back> complements the seating options. Adding warmth to the room, <8:a black cat> rests comfortably on the couch. Behind the sitting area, there is <5:a 4-layer wall-mounted wooden shelf> with additional decorative items, including <11,12: vases> and other decorative items, enhancing the cozy and inviting atmosphere. Closed to the shelf, there are several <9,19,22:potted plants with green leaves > are placed throughout the room, adding a touch of greenery. <2:The wall painted in warm tones>, create a cozy atmosphere and are adorned with framed artwork and decorations. <0: The floor is neutral-toned>, supporting the entire setup. The <3:ceiling painted white>, contrasts subtly with the walls and reflects the natural light entering the room through <6:the large windows>.

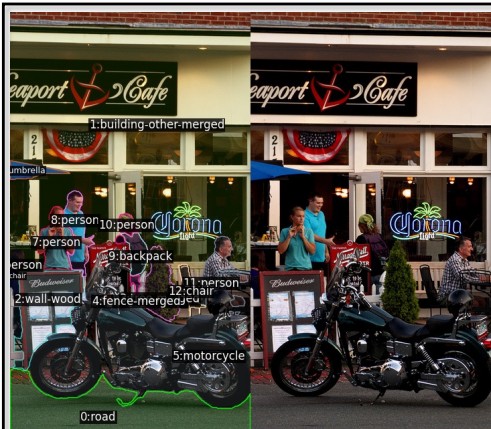

The image portrays a lively street scene outside a café. <0:The road> serves as the foreground, where <5:a motorcycle> is prominently parked, its shiny finishes and detailed designs drawing attention. Behind it, <1:the café building>, labeled as "Seaport Cafe" features large windows, a decorative sign, and patriotic bunting. A glowing neon "Corona Light" sign adds to the vibrant atmosphere. The café's exterior includes <2:a wooden wall> and <4:a small fence> which separates the outdoor seating area from the street. The seating area is equipped with <12,14:chairs> and <6:a blue umbrella> that provides shade for the patrons. <7,11: Several people are interacting in and around the café>, some standing while others are seated, enjoying their time. <10:One individual who is carrying <9:a black backpack> > is joining while there is <8:a guy in blue shirt> is trying to shake hands with her, adding to the dynamic social scene.

Figure 11: **Visualization of the Panoptic Grounded Caption.** Our annotated captions ground the panoptic segmentation masks.

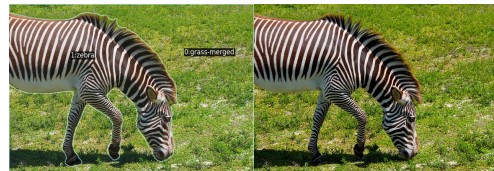

**COCONut-PanCap:** Positioned prominently within this image is <1: a zebra, easily distinguishable by its iconic black-and-white stripes and a striped mane along its neck>. Its belly is white, adding contrast to its overall pattern. The zebra is depicted in a natural feeding posture, with its head lowered as it grazes on <0:the grassland which is lush green>, forming the base of the scene.

**GPT-4V:** This image features a zebra grazing peacefully on a vibrant green field. The zebra's iconic black-and-white striped pattern stands out sharply against the lush grass, creating a striking visual contrast. Its head is lowered as it nibbles on the grass, showcasing its natural feeding behavior. The mane, with alternating dark and light stripes, adds a touch of elegance to its overall appearance. The bright sunlight enhances the vivid colors and intricate details, highlighting the beauty of the zebra in its natural habitat.

tier reason: Compared to COCONut-PanCap, GPT-4V shows more description for the behavior but add hallucination for sunlight.

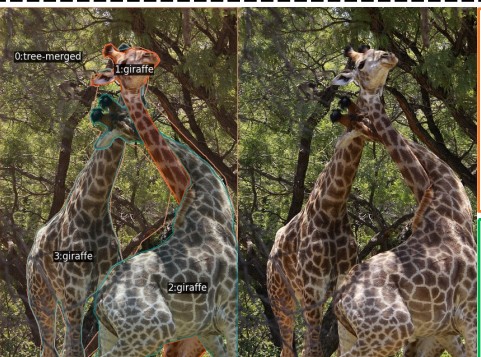

**COCONut-PanCap:** The image mainly features <0:the trees with very lush foliage>. In the front, there are <1,2: two giraffes with distinctive brown spots, yellow patterns, and long necks>. These giraffes are observed to be **fighting** by entwining their necks with each other. Behind them, there is <3: another giraffe, also with distinctive brown spots and yellow patterns>, which is standing still and partially obscured by the front giraffes.

**GPT-4V:** This image captures two giraffes engaging in a display of interaction, possibly sparring or playfully entwining their necks, a behavior commonly known as "necking." Set against a backdrop of lush green trees, the giraffes' distinctive spotted patterns stand out vividly in the sunlight filtering through the foliage. Their tall, elegant forms and graceful movements highlight the majestic nature of these animals, while the surrounding dense vegetation provides a natural and serene environment. The moment beautifully showcases the unique behavior and striking appearance of giraffes in their habitat.

tier reason: Compared to COCONut-PanCap, GPT-4V shows more description for the trees but add hallucination for the sunlight.

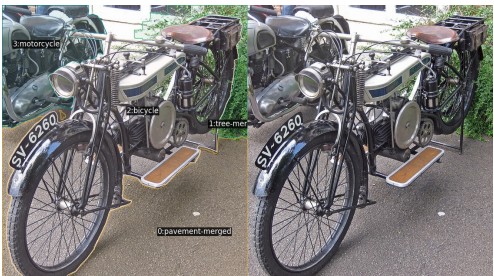

**COCONut-PanCap:** On <0:the gray asphalt sidewalk>, two motorcycles are parked side by side. <2:One motorcycle is predominantly black with white parts and a brown seat>, positioned to the left of <1:a green bush>, where a registration plate displaying the number "SV-6260" is visible to be mounted on the front fender, and a large cylindrical front headlamp is mounted on the front. <3:The other motorcycle is predominantly silver with a black seat>. The mechanical components include the single-cylinder engine and a straightforward exhaust system.

**GPT-4V:** This beautifully restored vintage motorcycle, likely from the early 20th century, showcases a classic design with a cream-colored fuel tank, spoked wheels, and a robust black frame. Key features include a large front headlamp, a registration plate reading "SV-6260," a brown leather saddle, and a small rear storage compartment. The visible single-cylinder engine and minimalistic exhaust system reflect the simple yet durable engineering of the era, with its pristine condition highlighting careful preservation.

tier reason: Compared to COCONut-PanCap, GPT-4V shows the story background, but missing details for region-3.

Figure 12: **Tier Examples for the User Study.** Our COCONut-PanCap annotations are tied with GPT-4V annotations for some simple cases.

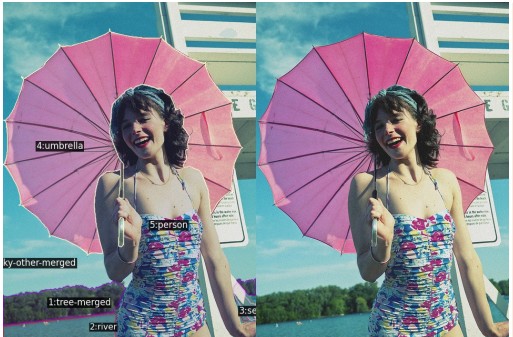

**COCONut-PanCap:** At the center of the scene, <5: a cheerful woman is wearing a vibrant swimsuit adorned with purple, blue, yellow, and white patterns>. She is holding <4:a pink umbrella>, which adds a pop of color and shade to the setting. The woman is smiling happily, radiating a joyful and carefree vibe, tying together the natural and human elements seamlessly. There are <1:lush green trees> and <0:blue sky> in the background, adding a natural element to the composition. Below it, there is <2: a serene blue river>, Together, the composition emphasizes the interaction between the lively figure and the tranquil outdoor environment.

**GPT-4V:** This vibrant image showcases a cheerful woman holding a bright pink parasol on a sunny day. She is dressed in a colorful floral swimsuit that exudes a retro style, complementing the joyful and carefree atmosphere. Her radiant smile and relaxed pose convey a sense of summer fun, with a backdrop of a clear blue sky, lush green trees, and a calm lake. The parasol adds a playful and vintage charm, enhancing the overall nostalgic feel of the scene.

tier reason: : Compared to COCONut-PanCap, GPT-4V recognize the water as lake, which is hard to be distinguished from this image.

- - - - - - - - - - - - - - - - - - - - - - - - - - - - - - - - - - - - - - - - - - - - -

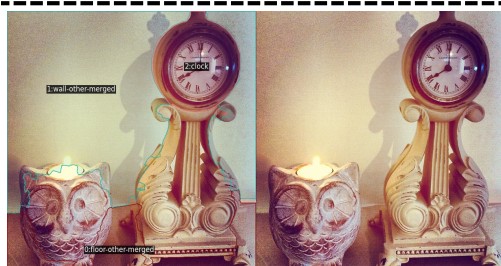

**COCONut-PanCap:** The image features an owl-shaped candlestick, and a clock are placed on <0:the floor>. The candlestick, designed in the form of an owl, adds a decorative element beside the clock. Prominently, in the center there is <2:a circular clock with hands and Roman numerals>, elegantly designed and placed on an intricate stand featuring ornate patterns and curves. Behind these items, there is <1: a plain and white wall>, providing a neutral background that highlights the objects in the foreground. Together, the objects and their placement create a cohesive and aesthetically pleasing composition.

**GPT-4V:** This cozy and artistic composition features a decorative vintage clock paired with a charming owl-shaped candleholder. The clock, with intricate scrollwork and Roman numerals on its face, exudes an antique elegance, while the soft glow of the candle creates a warm and inviting ambiance. The owl candleholder, with its textured surface and whimsical design, adds a playful yet rustic touch to the scene. The subtle lighting and shadows enhance the nostalgic and tranquil atmosphere, making it a perfect display for a quiet, reflective setting.

tier reason: Compared to COCONut-PanCap, GPT-4V summarize the style of the image as vintage.

- - - - - - - - - - - - - - - - - - - - - - - - - - - - - - - - - - - - - - - - - - - - -

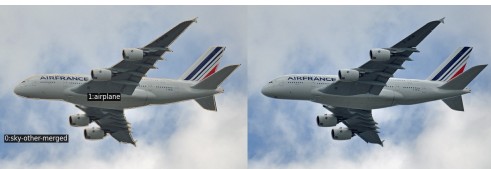

**COCONut-PanCap:** The image captures an airplane in flight. The center area highlights <1: an airplane, predominantly white in color, with dark blue English lettering "AIRFRANCE" on its body and distinctive dark blue and red patterns on its tail>. The airplane is depicted flying gracefully through the sky. The background is <0: the sky, characterized by a mix of blue open spaces and scattered clouds>.

**GPT-4V:** This image captures an Air France Airbus A380 in mid-flight against a backdrop of cloudy skies. The aircraft, known for its massive size and double-deck design, features the iconic Air France livery with its name prominently displayed on the fuselage and the red, white, and blue stripes on the tail, symbolizing the French flag. The four powerful engines are clearly visible, emphasizing the plane's engineering prowess. The angle of the shot highlights the sleek design and scale of the A380, showcasing its role as a leader in modern aviation.

tier reason: : Compared to COCONut-PanCap, GPT-4V summarize bit more details but with hallucination of model 'A380'.

Figure 13: **Tier Examples for the User Study.** Our COCONut-PanCap annotations are tied with GPT-4V annotations for some simple cases.

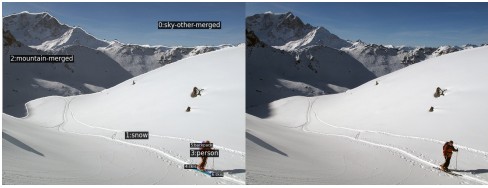

**COCONut-PanCap:** The image portrays a serene winter landscape. The image mainly highlights <0:a blue sky, creating a clear and bright backdrop>. Below it, there is <1:a pristine snow-covered slope, predominantly white and marked with ski trails> that curve across the terrain. Adjacent to this is <2:a majestic snow-covered mountain>, blending seamlessly into the snowy environment. At the center of the scene, <3:a person who is skiing down the slope, dressed in an orange jacket and black pants> is captured. The skier is equipped with skis attached to their feet, gliding smoothly over the snow. Additionally, <5:a backpack> is carried by the skier, completing their gear.

**GPT-4V:** This image captures a serene winter landscape featuring a skier ascending a pristine snow-covered slope. The untouched white snow is marked only by the skier's tracks and a winding trail that leads into the distance, emphasizing the tranquility and solitude of the scene. Majestic mountains with rugged, rocky peaks dominate the background, their shadows adding depth and contrast to the bright sunlight illuminating the snow. The skier, dressed in warm, vibrant gear and carrying poles, navigates the slope with determination, highlighting the spirit of exploration and adventure. The expansive sky above completes the breathtaking alpine setting.

tier reason: Compared to COCONut-PanCap, GPT-4V shows better vocabulary, but the information is almost the same.

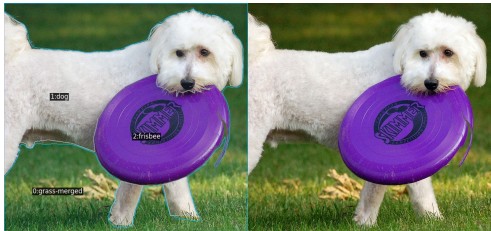

**COCONut-PanCap:** The image highlights <1:a white dog>, standing on the <0:lush green grass>, exuding a lively demeanor. The dog is holding <2:a purple frisbee> in its mouth. The frisbee features black text "SKIMMER" printed on its surface, adding contrast to its vibrant purple color. Together, the scene captures a moment of playful interaction, grounded by the mask numbers for clarity and detail.

**GPT-4V:** This image features an adorable white dog standing on a lush green lawn, holding a bright purple frisbee in its mouth. The dog's fluffy coat is well-groomed, and its expressive eyes gaze directly at the viewer, exuding charm and playfulness. The frisbee, labeled with "Skimmer Competition," is slightly chewed, indicating the dog's enthusiasm for play and activity. The vibrant colors of the frisbee and the green grass create a cheerful and lively atmosphere, perfectly capturing a moment of joy and companionship between a pet and its outdoor playtime.

tier reason: Compared to COCONut-PanCap, GPT-4V shows more OCR results of "competition", but the information is minor.

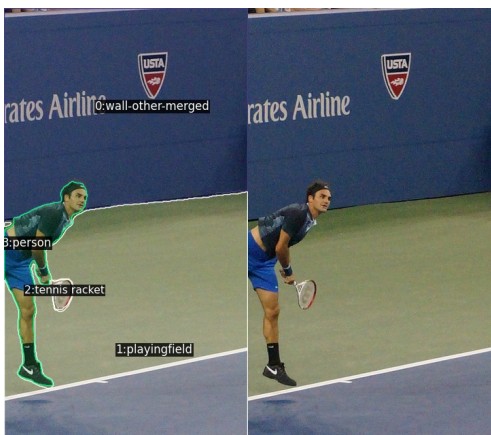

**COCONut-PanCap:** The image captures a dynamic tennis match. In the foreground, the image highlights <3:a man actively playing tennis>, dressed in a black top, blue pants, black socks, and black shoes. He is holding <2:a tennis racket>, characterized by its red and white frame and netting with printed letters. The background is <0:a predominantly blue wall>, decorated with red and black patterns along with white partial text "rates Airline". <1:The playing field is a mix of green and blue hues>.

**GPT-4V:** This image captures a tennis player intensely focused during a match, poised and ready to return a serve on a hard court. Dressed in a dark athletic shirt, blue shorts, and black sneakers, the player holds their racket firmly, leaning slightly forward in a balanced stance that conveys readiness and determination. The backdrop prominently features the USTA logo and the Emirates Airline sponsorship, indicating a professional tennis tournament setting. The court's clean lines highlight the competitive atmosphere

tier reason: Compared to COCONut-PanCap, GPT-4V shows extra wording to describe the atmosphere which is not necessary.

Figure 14: **Tier Examples for the User Study.** Our COCONut-PanCap annotations are tied with GPT-4V annotations for some simple cases.

# E  Dataset Information

The proposed dataset COCONut-PanCap builds on top of COCO [36] and COCONut [9]. We provide their datast information below:

**COCO**: The COCO2017 dataset [36] contains 118K images for training, and 5K images for validation.

License: https://cocodataset.org/termsofuse

URL: https://cocodataset.org

**COCONut**: The COCONut [9] dataset modernizes COCO segmentation dataset. We mainly use their COCONut-S and COCONut-val splits.

License: https://github.com/bytedance/coconut_cvpr2024/blob/main/LICENSE

URL: https://xdeng7.github.io/coconut.github.io/

