# OpenReview forum: "COCONut-PanCap: Joint Panoptic Segmentation and Grounded Captions for Fine-Grained Understanding and Generation"
_NeurIPS.cc/2025/Datasets_and_Benchmarks_Track — NeurIPS 2025 Datasets and Benchmarks Track poster_

### Official Review · Reviewer_n5Bx · 2025-06-03

**Rating:** 4
**Confidence:** 5

**Summary:**

This paper proposed a caption dataset **COCONut-PanCap** combining panoptic segmentation and grounded captioning. It comprises 118K image- text pairs for training, with an average caption length of 203 words, as well as an additional 25K image- text pairs, with an average caption length of 233 words for validation. This  dataset provide high quality detailed caption and instance masks in the era of detailed captioning. And the paper conduct the experiments on three primary tasks: detailed caption generation, panoptic segmentation and grounded captioning, text-conditioned image generation and two down-stream task visual question answering, and referring segmentation. The model trained on COCONut-PanCap achieve SOTA performance on each task, validate the dataset's quality.

**Additional Feedback:**

Questions

1. The avg money / time cost for each annotation is not discussed, as saying in the limitation. High-quality human-labeled data offers significant benefits for instruction tuning in multi-modal tasks, but scaling such datasets is challenging. I want to know the actual of cost of building such a dataset.

**Dataset Code Accessibility:**

Partly

**Dataset Code Comments:**

Only part of annotation (50K) is avaliable on github.

**Ethical Considerations:**

No, there are no or only very minor ethics concerns

**Final Justification:**

Authors' detailed response has successfully addressed most of my previous concerns.

My primary remaining concern revolves around the choice of the dataset. While the COCO dataset has been foundational for the field, its scenes can be considered relatively simple compared to the complexity encountered in many real-world applications today. I believe a significant next step for this research area would be to curate and annotate new, more challenging datasets. These datasets, featuring greater scene complexity or more diverse domains, would not only serve as a training dataset but could also spur the development of the next generation of creative image annotation pipeline. I'd like to keep my rating.

**Limitations Weaknesses:**

1. Dataset Scale & Resolution:​​ While MSCOCO is a pragmatic foundation, its resolution and category diversity limit T2I generation quality and potential for high-resolution MLLMs. The annotation pipeline could be highly impactful if applied to larger-scale datasets like ​​SA-1B​​, despite its lack of captions. ​

2. Quantify Coverage Metrics:​​ The absence of quantitative ​​image coverage rate​​ metrics makes it difficult to compare COCONut-PanCap's grounding completeness (e.g., Figure 13 shows uncovered instances like the candlestick). Report and compare coverage rates (e.g., % of image area covered by masks)

3. T2I Training:​​ Using only LoRA fine-tuning for T2I experiments may underutilize the dataset's potential. High-quality, dense datasets are often most effective for ​​full-model fine-tuning or continual pre-training​​, potentially leading to greater quality leaps.

4. The result in Figure 7 is not convergent: it only show the case of rasing human annotated data ratio, not show the tendency of decreasing synthetic captions ratio. A direct question is: only using human annotated data is the best case?

5. Lack of result visualization of success example and faliure cases for each task.

6. The quality of the annotation is uneven. There are some bad case, e.g, <Obstruction> for coconut_pancap_50k/000000000626. In the first version, seemly the dataset are first annotated in Chinese, then transfer to english. Some caption even include chinese.

**Strengths Contributions:**

​High-Quality Dataset:​​ COCONut-PanCap significantly advances detailed captioning datasets by providing both densely detailed captions and comprehensive instance masks—a valuable pairing absent in prior datasets like GranD. This bridges a critical gap for training multimodal models requiring precise cross-modal alignment.

Rigorous Multi-Task Validation:​​ The extensive experiments across five diverse tasks (including captioning, segmentation, VQA, and T2I generation) robustly demonstrate the dataset's versatility and effectiveness in enabling SOTA performance. This breadth strongly validates the dataset's quality.

The writing is clear and the chart is beautiful.

---

> ### Author Rebuttal · Authors · 2025-07-31
>
> We thank the reviewer for the constructive feedback and carefully address the concerns below.
>
> **W1. Dataset Scale & Resolution**
>
> We thank the reviewer for the suggestion and we totally agree with it. We could leverage our annotation pipeline to enrich more image sources in our future work.
> Currently, we have already enriched the COCONut-PanCap validation set by using additional 20K images from the validation set of Objects365 (which provides denser objects than the original COCO validation set).
>
> **W2. Quantify Coverage Metrics:​​ grounding completeness.**
>
> Even though we have tried our best to cover as many regions as possible, we agree with the reviewer that some small area remains uncovered but this is a natural issue when creating dense captions, not every pixel can be covered by description. But our annotations have been covered 98% area per image.
>
> **W3. T2I Training:​​  effectiveness for ​​full-model fine-tuning.**
>
> Thanks for pointing it out. We conduct fully finetuning for SD3 in the same settings. Our experimental results show that fully fine-tuning achieves better performance of which GenEval score is 0.71 compared to the previous lora finetuning setting where we only achieve 0.68 GenEval score. When we use the original COCO caption to fully finetune the model, it achieves a similar score of 0.66. We will add this experimental result in our camera ready.
>
> **W4. The result in Figure 7 is not convergent: only using human annotated data is the best case?**
>
> We would like to clarify the experimental setting in detail and further elaborate on the conclusion. In this study, we investigate the impact of varying proportions of human-annotated data on model performance while keeping the total data volume constant. Specifically, we fix the total number of training examples and gradually replace the synthetic annotations with human-annotated ones in increments (e.g., 0%, 25%, 50%, 75%, and 100%). Our results show that performance improves consistently as the proportion of human-annotated data increases, with the best results achieved when 100% of the training data is human-annotated.
>
> We agree with the reviewer’s observation that the trend in the figure has not fully converged, which in fact supports the idea that adding more human-annotated data could further enhance performance. This aligns with our broader hypothesis that the quality and grounding precision of human annotations are critical for training robust models.
>
> **W5. Lack of result visualization of success example and faliure cases for each task.**
>
> Thanks for the valuable comment. One of the failure cases is to caption and segment challenging objects, such as small/thin or heavily occluded objects. Due to the rebuttal constraint that we could not provide visualizations in rebuttal, we will instead add them in the revision.
>
>
> **W6. There are some bad cases.**
>
> Thanks for the careful check. We have multiple revisions on the dataset to maintain the quality, we will fix the translation problem in our public release.
>
> **Q1. Annotation cost.**
>
> Thanks for pointing it out. We will introduce it in the limitation in our appendix for camera-ready.  In detail, the costs depend on the number of edits per image.  For the number of edits less than 5 (easy split), each image takes 0.42 USD. For the number of edits more than 5 (hard split), each image takes 0.98 USD. For all images 118K (train) + 25K (val), there are 65K images for the easy split and 78K images for hard split. Therefore, it results in the total cost of 27.3K + 76.4K = 103.7K USD. It is noted that the average cost includes all the GPT4V API fee, human editing fee, and human verification fee.
>
> **C1. Only part of annotation (50K) is available on github.**
>
> There is another link for the full dataset appended in the submission

---

> > ### Comment · Reviewer_n5Bx · 2025-08-05
> >
> > Thanks for addressing my concerns. Given annotations have covered 98% area per image and enriched the COCONut-PanCap validation set, I'll not lowering my score. I support acceptance.

---

### Official Review · Reviewer_qbH7 · 2025-06-19

**Rating:** 5
**Confidence:** 3

**Summary:**

The paper introduces COCONut-PanCap, a dataset designed to improve panoptic segmentation and grounded image captioning. It is built on the COCO dataset and recent COCONut panoptic segmentation annotations. The authors assembled 143,000 image-text pairs (118k train / 25k val), each featuring "long-form captions" (avg. 200+ words) grounded in panoptic segmentation masks.

The dataset is built using a hybrid pipeline: region-level captions are generated with GPT-4V and then curated by humans. These region-based descriptions are aggregated into image-level captions that preserve "grounding" to the underlying segmentation masks.

COCONut-PanCap is tested for text-to-image generation, visual question answering (VQA), and the most novel task: panoptic segmentation grounded caption, which generates text captions along with segmentation masks.

**Dataset Code Accessibility:**

Yes

**Dataset Code Comments:**

The dataset has been uploaded to HuggingFace. It includes a short REAME to download the corresponding COCO images and COCO-Nut S panoptic masks.

The code to reproduce experiments is not public (see Weaknesses).

**Ethical Comments:**

COCO images have been widely used for years. This study only provides additional captions, so I don't see any ethical concerns.

**Ethical Considerations:**

No, there are no or only very minor ethics concerns

**Final Justification:**

I recommend acceptance. Authors have addressed points raised during rebuttal. I keep my confidence to 3 as I am not an expert in the field.

**Limitations Weaknesses:**

-  Most evaluations are centered around captioning, segmentation, and generation using the same dataset. My main concern is about overfitting to the benchmark. It would strengthen the work to include evaluation on external datasets.

- While the original COCONut segmentations included 133 classes, the authors claims to introduce additional concepts here. However, the authors could more explicitly detail the nature and extent of this expanded concept diversity (see Section 3.3, "Concepts Beyond COCO’s 133 Classes.").

-  The paper does not discuss where the models/datasets underperform. Including qualitative or quantitative failure cases (e.g., on small/thin objects, occlusion, uncommon object combinations) would help.

- While the paper studies synthetic vs. human-annotated captions, which is interesting, it does not isolate the impact of human correction of GPT outputs versus fully manual captions. In other words, what are the annotation cost vs. benefit trade-offs?

- The code to reproduce experiments doesn't seem to be public yet. Will it be made public upon acceptance?

**Strengths Contributions:**

- The paper is clearly written and easy to follow.  To my knowledge, the related work is well covered. The Figures and Tables are clear.

- The generated captions are of high quality. The results of the reader study clearly show that the caption quality is better than that produced by GPT-4V. The included examples are also convincing. The idea of transforming existing segmentation annotations into grounded textual descriptions is both intuitive and compelling.

- The paper introduces the new task of Panoptic-Grounded Captioning (PGC), along with the right dataset to train it. This task combines panoptic segmentation and grounded image captioning in a single framework. While both tasks have been studied on their own, this is, to my knowledge, the first large-scale effort to bring them together using detailed pixel-level masks and carefully edited scene descriptions.

-  The ablation on synthetic vs. human-annotated data is a useful contribution. The authors show that even a small amount of human supervision (e.g., 25%) leads to high gains in both captioning and generation metrics. This highlights the limits of fully synthetic training, which is often assumed to be sufficient in large-scale vision-language settings.

---

> ### Author Rebuttal · Authors · 2025-07-31
>
> We thank the reviewer for the constructive feedback and carefully address the concerns below.
>
> **W1. Overfitting to the benchmark and include evaluation on external datasets.**
>
> Thanks for the suggestion. In the submission, we have conducted diverse experiments to verify the data power of our proposed dataset. Specifically, for text to image generation, we verified the model’s effectiveness by training it on our dataset and evaluating it on the GenEval and DOCCI test set as shown in Tab. 5 and Tab. 6. Additionally, Tab. 7 and Tab. 8 also provided results evaluated on various VQA benchmarks, and referring segmentation tasks.
>
> **W2. "Concepts Beyond COCO’s 133 Classes."**
>
> We thank the reviewer for pointing this out. However, as analyzed in our paper, the annotations in fact extend beyond the 133 classes and have been shown to work effectively for vision-language models (VLMs). Specifically, we leverage the masks of 133 classes to create the grounded caption but it does not mean that we limit the categories within the 133 classes. For example, as shown in our appendix,  in the last row example in Fig.10, mask_3 is described as green lush where in COCO it was instead labeled as tree. Tree is a very coarse concept but in our annotation the concept is more fine-grained.  Additionally, We found that with our detailed captions as additional instruction data, VLM model is able to improve tasks with various out-of-domain categories such as the MMBench benchmark (Tab. 7), thanks to the strong generalizability of modern VLMs. This indicates the limit on the object categories may be a minor issue for instruction-tuning.
>
>
> **W3. Models/datasets underperform. Including qualitative or quantitative failure cases**
>
> We thank the reviewer for the suggestion. Indeed, our current models fail to caption challenging objects, such as small/thin or heavily occluded objects. For example, as shown in the second row of Fig.10, there is a metal fence in front of the objects, which is not labeled with any mask or caption. This is a challenging case, as it covers a very small region and is very thin to be segmented. We note that it is not a significant failure case for the dense understanding of the whole image. Finally, due to the constraint in the rebuttal, we cannot provide the visualizations of more failure cases. Nevertheless, we will add them along with discussions in the revision.
>
> **W4. What are the annotation cost vs. benefit trade-offs?**
>
> Thanks for pointing it out. We will introduce it in the limitation in our appendix in camera ready.  In detail, the costs depend on the number of edits per image.  For the number of edits less than 5 (easy split), each image takes 0.42 USD. For the number of edits more than 5 (hard split), each image takes 0.98 USD. For all images 118K (train) + 25K (val), there are 65K images for the easy split and 78K images for hard split. Therefore, it results in the total cost of 27.3K + 76.4K = 103.7K USD. It is noted that the average cost includes all the GPT4V API fee, human editing fee, and human verification fee. One of the benefit trade-offs could be leveraging the high-quality annotated data to train a better model to generate better quality synthetic data in order to deliver large scale high-quality dataset.
>
> **W5. The code to reproduce experiments**
>
> Thanks for the comment. As promised in the submission, we will make the whole dataset along with code publicly available upon acceptance.

---

> > ### Comment · Reviewer_qbH7 · 2025-08-05
> > **Post rebuttal**
> >
> > Thanks for addressing my points. Assuming the final version will include a discussion of failure cases with public code to reproduce the whole evaluation, I support acceptance.

---

### Official Review · Reviewer_P166 · 2025-07-01

**Rating:** 4
**Confidence:** 4

**Summary:**

This paper presents the COCONut-PanCap dataset, designed to enhance both panoptic segmentation and grounded image captioning. Building upon the COCO dataset, this new dataset features advanced COCONut panoptic masks and aims to address the shortcomings of existing image-text datasets, which often provide inadequate and less detailed scene descriptions. The COCONut-PanCap dataset includes fine-grained, region-level captions that are aligned with the panoptic segmentation masks, ensuring consistency and improving the quality of generated captions. With human-edited, densely annotated descriptions, COCONut-PanCap facilitates enhanced training for vision-language models (VLMs) in image understanding and supports generative models for text-to-image tasks.

**Dataset Code Accessibility:**

Yes

**Ethical Considerations:**

No, there are no or only very minor ethics concerns

**Final Justification:**

Thank you for the rebuttal. Most of my concerns are addressed. Therefore, I will adjust my rating.

**Limitations Weaknesses:**

- The novelty of the proposed dataset might be incremental. Compared to the GLaMM-GCG dataset, the only apparent difference appears to be the large number of masks, while the task remains the same and the image domain is similar. It remains unclear what unique insights or new capabilities this dataset offers.
- The annotation pipeline heavily depends on human input, raising concerns about scalability. Additionally, the paper does not clearly demonstrate whether the added annotation precision significantly benefits downstream models more than training on larger, noisier datasets. An experimental comparison (e.g., between LLaVA 665K-COCONut-PanCap-20K and LLaVA 665K-ShareGPT4V-100K) would help clarify this.
- The proposed dataset is built upon a limited class list of the COCO dataset (133 classes) compared to other existing large-scale datasets, which limits the diversity of the dataset. Is there a solution to deal with out-of-distribution categories?
- The paper lacks a detailed explanation of design choices for the dataset construction. For example, how are the prompts used for each round decided? What if we have more than 2 rounds of annotations?

**Strengths Contributions:**

-  The authors constructed a new large-scale dataset.
- The authors utilized GPT4 to improve the efficiency of labeling a segmentation dataset with caption labels.

---

> ### Author Rebuttal · Authors · 2025-07-30
>
> We thank the reviewer for the constructive feedback and carefully address the concerns below.
>
> **W1. Comparison of GLaMM-GCG**
>
> GLaMM-GCG dataset leverages automated pipeline to ground the captions with an average mask number  of 3.6. However, ours is much denser with an average mask number of 13.2, providing the benefit of dense pixel understanding. Recently, pixel-level understanding for vision language models requires dense captions grounded with masks as shown in recent study of DenseWorld-1M. GLaMM-GCG yields sparse grounded caption which does not meet the expectation for the future study. Besides, our dataset has longer average words per caption of 203 compared to GLaMM-GCG of which the average words is only 104.
>
> Reference:
>
> [A] DenseWorld-1M: Towards Detailed Dense Grounded Caption in the Real World, Xiangtai, Li, etc. arxiv, 2025
>
>
> **W2. Human annotated data VS VLM annotated data.**
>
> ShareGPT4V-7B was trained on LLaVA1.5 with language model of Vicuna-7B using “the original 665K instruction tuning data with the replaced 23K detailed descriptions randomly sampled from ShareGPT4V-100K.” We follow similar settings to replace the original 23K detailed captions with COCONut-PanCap-20K or ShareGPT4V-100K to train models individually. We observe that using only 20K human labeled data can still achieve comparable performance to 100K synthetic data.
>
> | Training data           | SEED | MME  | MMBench | TextVQA | MM-Vet |
> |------------------------|------|------|---------|---------|--------|
> | ShareGPT4V-100K        | 67.4 | 1519 | 70.5    | **64.6** | 37.8   |
> | COCONut-PanCap-20K     | **67.7** | **1521** | **70.9** | 64.5    | **38.5** |
>
>
> **W3. Limited class list of the COCO dataset (133 classes).**
>
> We thank the reviewer for pointing this out. However, as analyzed in our paper, the annotations in fact extend beyond the 133 classes and have been shown to work effectively for vision-language models (VLMs). Specifically, we leverage the masks of 133 classes to create the grounded caption but it does not mean we limit the categories within the 133 classes. For example, as shown in our appendix,  in the last row example in Fig.10, mask_3 is described as green lush where in COCO it was instead labeled as tree. Tree is a very coarse concept but in our annotation the concept is more fine-grained.  Additionally, We found that with our detailed captions as additional instruction data, VLM model is able to improve tasks with various out-of-domain categories such as the MMBench benchmark (Tab. 7), thanks to the strong generalizability of modern VLMs. This indicates the limit on the object categories may be a minor issue for instruction-tuning.
>
>
> **W4. Details of prompt template.**
>
> The prompt template remains the same, and the only difference is the human effort leverage. Human raters decide the data to go through more rounds of editing when they consider the data annotations do not meet the human-level quality.

---

> ### Comment · Reviewer_P166 · 2025-08-05
>
> Thank you for the rebuttal. Most of my concerns are addressed. Therefore, I will adjust my rating.

---

### Official Review · Reviewer_sTMr · 2025-07-02

**Rating:** 5
**Confidence:** 4

**Summary:**

This paper introduces COCONut-PanCap, a high-quality image-text dataset designed for vision-language tasks such as understanding, generation, and segmentation. To create this dataset, the authors developed a sophisticated annotation pipeline. In this pipeline, images with human-annotated masks are used to prompt a Vision-Language Model (VLM), which generates initial detailed captions. These captions are then manually refined and subsequently summarized by another VLM, followed by a final human verification step. Leveraging this dataset, the authors conducted a diverse set of experiments to evaluate its performance on understanding, generation, and segmentation tasks. The experimental results demonstrate that the dataset can remarkably improve performance across these different applications.

**Additional Feedback:**

NA

**Dataset Code Accessibility:**

Yes

**Ethical Considerations:**

No, there are no or only very minor ethics concerns

**Final Justification:**

I have read the authors' response and believe this paper is acceptable in Neurips.

**Limitations Weaknesses:**

- To better validate the efficacy of COCONut-PanCap, a direct comparison is recommended. This would involve finetuning a model like SD3 on a version of the COCO dataset captioned solely by a VLM (without manual refinement) and comparing its performance against the same model finetuned on COCONut-PanCap.
- The reliance on manual human supervision limits the scalability of the annotation pipeline, making it challenging to produce datasets on a significantly larger scale.

**Strengths Contributions:**

- The paper is well-written and clearly presented. The core idea of linking panoptic segmentation with detailed captions and referring segmentation is both novel and compelling.
- The proposed annotation pipeline is well-designed and effective. By incorporating human supervision, this pipeline successfully generates a high-quality dataset with dense, detailed captions.
- The resulting COCONut-PanCap dataset demonstrates impressive performance across various tasks, including segmentation, understanding, and generatio

---

> ### Author Rebuttal · Authors · 2025-07-30
>
> We thank the reviewer for the constructive feedback and carefully address the concerns below.
>
> **W1. A direct comparison between COCO solely recaptioned by VLM and COCONut-Pancap**
>
> To enable a direct comparison between captions generated solely by a vision-language model (VLM) and our COCONut-PanCap dataset, we recaptioned the COCO dataset using GPT-4V, resulting in COCO-GPT4V. GPT4V has been verified by human evaluation to produce high-quality synthetic captions as shown in Fig.6. We then trained SD3 on COCO-GPT4V under the same settings to make a fair comparison. The resulting GenEval score for COCO-GPT4V was 0.65, which does not perform as well as our COCONut-PanCap dataset of which the score was 0.68. This supports our motivation: human effort can still enhance data quality, indicating that there is room for improving synthetic data.
>
> **W2. The reliance on manual human supervision limits the scalability of the annotation pipeline, making it challenging to produce datasets on a significantly larger scale.**
>
> We realized the challenge but still resort to human labeling to ensure the high-quality of data for instruction tuning. The goal of our paper is to create a high quality human annotated dataset for the future work of scaling up the dataset with more diverse image sources. We conduct extensive experiments to compare the synthetic and human annotated data. Our experimental results show that our high-quality data help boost performance for various tasks. We are currently extending the human annotated dataset to a larger scale and will try to release it in the future. Again, as discussed in the limitation in the appendix,  high-quality caption data is rare in multi-modal learning, we believe COCONut-PanCap serves as an initial step toward scalable high-quality detailed captioning datasets for training seed models for large-scale synthetic data (but beyond the scope of this submission). Besides, it is essential to use human-annotated data in existing popular VLM evaluation benchmarks like TextVQA, MME, and MMBench.

---

### Note · Authors · 2025-08-14

We sincerely thank all reviewers for their constructive feedback and unanimously positive initial scores. We are pleased that our rebuttal effectively addressed the reviewers’ concerns, and that during the final discussion stage, all reviewers continued to lean toward acceptance. We also extend our gratitude to the AC for coordinating the review process and facilitating constructive discussions.

In this work, we present COCONut-PanCap, a high-quality image-text dataset tailored for diverse vision-language tasks, including understanding, generation, and segmentation. The dataset’s quality is ensured through a carefully designed annotation pipeline. Experimental results demonstrate that COCONut-PanCap yields improvements across multiple tasks, underscoring its value to the community.

We appreciate the thoughtful suggestions provided and will incorporate the recommended improvements in the final version. We believe COCONut-PanCap will serve as a valuable resource for advancing multi-modal learning research.

---

### Decision · Program_Chairs · 2025-09-18

**Decision:**

Accept (poster)

**Comment:**

All reviewers recommend acceptance, and I agree with this consensus. The paper presents a strong and well-executed contribution that will be of clear value to the community. The methodology is sound, the experiments are thorough, and the results convincingly demonstrate the effectiveness of the proposed approach.

Despite the concern of the choice of dataset, the paper makes a strong contribution and meets the standards of a top-tier conference. I recommend acceptance